# Physiological and Molecular Responses to Acid Rain Stress in Plants and the Impact of Melatonin, Glutathione and Silicon in the Amendment of Plant Acid Rain Stress

**DOI:** 10.3390/molecules26040862

**Published:** 2021-02-06

**Authors:** Biswojit Debnath, Ashim Sikdar, Shahidul Islam, Kamrul Hasan, Min Li, Dongliang Qiu

**Affiliations:** 1College of Horticulture, Fujian Agriculture and Forestry University, Fuzhou 350002, China; biswo26765@yahoo.com (B.D.); liminzyl@sina.com (M.L.); 2Department of Horticulture, Sylhet Agricultural University, Sylhet 3100, Bangladesh; shahidul.hrt@sau.ac.bd; 3Department of Agroforestry and Environmental Science, Sylhet Agricultural University, Sylhet 3100, Bangladesh; ashim.aes@sau.ac.bd; 4Department of Agricultural Chemistry, Sylhet Agricultural University, Sylhet 3100, Bangladesh; hasanmk.agrichem@sau.ac.bd

**Keywords:** acid rain, oxidative stress, antioxidant activity, silicon, glutathione, melatonin

## Abstract

Air pollution has been a long-term problem, especially in urban areas, that eventually accelerates the formation of acid rain (AR), but recently it has emerged as a serious environmental issue worldwide owing to industrial and economic growth, and it is also considered a major abiotic stress to agriculture. Evidence showed that AR exerts harmful effects in plants, especially on growth, photosynthetic activities, antioxidant activities and molecular changes. Effectiveness of several bio-regulators has been tested so far to arbitrate various physiological, biochemical and molecular processes in plants under different diverse sorts of environmental stresses. In the current review, we showed that silicon (tetravalent metalloid and semi-conductor), glutathione (free thiol tripeptide) and melatonin (an indoleamine low molecular weight molecule) act as influential growth regulators, bio-stimulators and antioxidants, which improve plant growth potential, photosynthesis spontaneity, redox-balance and the antioxidant defense system through quenching of reactive oxygen species (ROS) directly and/or indirectly under AR stress conditions. However, earlier research findings, together with current progresses, would facilitate the future research advancements as well as the adoption of new approaches in attenuating the consequence of AR stress on crops, and might have prospective repercussions in escalating crop farming where AR is a restraining factor.

## 1. Introduction

Societies have been using numerous natural means for their existence since the beginning of civilization. By using many of the earth’s energy sources, people have made their lives easier. Contrarily, it has created pollution due to the released hazardous materials to the environment. Fossil fuel combustion from vehicles, industrial flourish and urbanization have increased the concentration of fumy and particulate impurities in the atmosphere, which causes air pollution [1]. Acid rain (AR) occurs due to intensive air pollution and it has become a common phenomenon worldwide, especially in Europe, East Asia and North America [2]. In China, AR has been documented as one of the major environmental pollution factors in recent years owing to the increasing commercial development [3,4]. The dissemination of AR was found to be increased in China since the 1970s and the occurrence of AR has predominantly been reported in Southern China. It was reported that only 11 provinces of South China experienced the loss of ecological benefits of more than US$2.4 billion annually because of AR [5]. Therefore, the soil of a vast area in these parts has been found to be acidified and thereby major ecosystems are also in a vulnerable condition [4].

Robert Angus Smith was a pharmacist of Manchester, England, who observed high levels of acidity in rain water over industrial zones of England in the year of 1852 and discovered the phenomenon of AR [6]. In contrast, he observed lower acidity levels in rain water of slightly polluted areas, particularly near the coast [7]. Until the 1950s, his work was not able to arrest public attention. But, when biologists reported a drastic drop in the fish population of the southern Norway lakes as well as harmful effects on vegetation due to acidic rain water, scientists began to focus on the details of AR, like the way of its formation, nature and possible intensity of occurrence, and its impact on the earth [7].

Currently, AR is considered as a potential threat to the agriculture sector. Scientists piloted their researches through simulated acid rains (SAR) that may change the growth, development, physiological and molecular activities of plants, as well as a decline in the output [8]. AR impedes the basic plant growth indicators like plant height, leaf number, diameter of stem and shoot and root fresh biomass, implying that SAR stress in plants causes reduced plant growth and development [9,10,11]. In addition, AR hampers the photosynthetic activity of plants and thus, decreases the photosynthetic rate. Besides, the level of alteration in plant photosynthetic activities because of AR condition varied among plant species and with the level of stress [12,13]. Basically, AR deposition affects the ultrastructure of chloroplast and leaf plasma membrane, which result in lower photosynthetic activity and degradation of chlorophyll [14]. Ultimately, AR stress causes the accumulation of reactive oxygen species (ROS) and melondialdehyde (MDA) contents in plant cells [8,12,13]. It was observed in different crops that the antioxidant defense system was strengthened in response to moderate AR stress conditions to scavenge ROS and reduce oxidative injury, but the ROS detoxifying ability decline in severe stress conditions might be due to the changes in metabolic status or their biosynthesis [8]. Furthermore, AR causes the alternation in the differentially expressed genes and transcriptional factors [15]. In the past decades, the mechanisms of plant response to AR stress were very briefly elucidated in tomato and research on AR stress mitigation processes were not done in tomato.

It has been studied that different bioactive compounds such as silicon, glutathione and melatonin can improve the abiotic stress tolerance, including AR stress in plants [16,17,18]. Silicon is the second richest element in the soil and acts as a useful component for higher plants [19]. It has been established that silicon can improve plant tolerance against different abiotic stresses including salinity, drought, metal toxicity, etc., and biotic stresses including pathogens and insects [19,20,21,22,23].

Similarly, glutathione is considered one of the major non-protein thiol bioactive water-soluble compounds (Phytochelatin) in plant cells and plays diverse biochemical roles in plants to adapt to abiotic stresses through ROS scavenging directly and/or activating different antioxidant compounds [24,25]. The phytochelatin also plays a vital role in protecting cellular functions through different mechanisms and metal/metalloid homeostasis by performing their chelation and/or detoxification [26]. In addition, glutathione plays a vital role in growth, development, photosynthetic activity, expression of differential genes and activation of protein by means of its diverse properties under abiotic stress conditions [17,24,25,27].

On the other hand, melatonin (*N*-acetyl-5-methoxytryptamine) is a low molecular weight natural molecule which is present in living organisms, spreading from mammals to bacteria [28]. It is well-reported that melatonin has important positive functions in animal and plant physiology as well as in various human processes [29,30,31,32]. The pleiotropic biological activities of melatonin in living organisms are arbitrated by membrane receptors and nuclear receptors [33,34]. Moreover, melatonin receptors work independently [35], and their bioactive metabolites affect the interactions of melatonin with ROS [36].

As it is amphiphilic in nature, melatonin can easily infiltrate the cell membrane and dispense to the cytosol, the nucleus and mitochondria [37]. In fact, melatonin plays a crucial role in the non-receptor-mediated activities such as quenching ROS, and enhancing antioxidant capacity, protecting living cells from oxidative injury [38,39,40]. Consequently, the formation and absorption of ROS are the elementary processes related to cellular biology and physiopathology. Thereby, it is anticipated that the principal role of melatonin in living organisms is to strengthen the antioxidant system and act as a front-line defense against any adverse environment [41].

In the current review, emphasis has been given on the systematic and deep exploration of the recent advances in research of AR stress mitigation in plants by the supplementation of different bioactive compounds like silicon, glutathione and melatonin. Notably, the mechanisms of bioactive compound-mediated AR stress tolerance in plants have gradually been exposed. Therefore, the effects of AR on the photosynthesis potential, metabolic mechanism of ROS accumulation and mechanism of AR stress tolerance-related gene expression in plants have been summarized. However, based on the previous and current research outputs, the detailed impact of exogenous silicon, glutathione and melatonin on plant metabolic processes underlying the AR stress tolerance mechanism has been elucidated, which would be perceptive and supportive for the future research in AR stress amelioration using the studied bioactive compounds as well as might pave the way for attempting new compounds in different plants.

## 2. Impacts of Acid Rain on the Photosynthesis Potential in Plants

Photosynthesis is a fundamental physio-morphological process to sustain plant life activities, including growth and development, and this process helps in the synthesis of organic composites by the usage of light energy in plants [17,42]. Photosynthetic pigments like chlorophyll and carotenoids are vital for photosynthesis, which is essential for the growth and development of plants, and these are the penetrating signs to observe the harmful impacts of different environmental hazards on germination, seedling growth, leaf structure, health and function [43,44,45]. Chlorophyll converts carbon-dioxide and water into biochemical energy like carbohydrates and oxygen by using light energy [46]. Basically, the chlorophyll content in plants indicates the effectiveness of photosynthesis, and the increased ROS accumulation in plants under environmental stress conditions triggers a significant decline of chlorophyll content in plant leaves because of its fragile nature [47]. Contrarily, carotenoids act as a safeguard for the photosynthetic apparatus by quenching ROS through the xanthophyll cycle under biotic and abiotic stress conditions [48].

Currently, it has been observed in several experiments that chlorophyll concentration as well as photosynthesis in tomato plant are severely affected by environmental stresses, including high temperature, low temperature, salinity, alkalinity, drought, metal toxicity and others [49,50,51,52,53,54]. AR is also considered a major abiotic hazard due to its hostile influence on bio-energetic advancement of photosynthesis [55,56]. Previous research reported that the chlorophyll content significantly declined due to AR in tomatoes [57] and the trend of declination was associated with the extent of AR stress and plant species [58]. The authors observed that the lessening of chlorophyll a and chlorophyll b content was higher at pH 2.5 of AR compared to pH 3.5 of AR in leaves of two different types of tomato cultivar, namely Micro-Tom as a determinate type and Red Rain as an indeterminate type. Moreover, the degradation of chlorophyll concentration is a distinctive indicator of leaf senescence, which causes yellowing of leaves in tomato plant [58]. It has been stated that chlorophyll loss is accompanied with the upregulation of chlorophyll degradation genes, for example *SGR1* and *PAO* [59,60]. On the other hand, carotenoid contents in tomato leaves decrease markedly under AR stress and the decrease level depends on pH level of AR water [58]. The results indicate that AR, having a higher acidity level, causes a higher amount of light absorbance, which releases maximum heat, resulting in lower usage of light energy and damage to photosynthesis pigments [44]. Therefore, the growth, development and survival of plants depend on functional photosynthesis apparatus, which are fundamentally hampered by AR stress.

Chlorophyll fluorescence is certainly affected by abiotic stresses and commonly used as a sensitive indicator to observe the photosynthetic status of plants [61]. The measurement of chlorophyll fluorescence is considered as a rapid, prominent and reliable technique to evaluate photosynthetic activities in any stressed plant [43,62]. The measurements of the primary photosystem II (PSII) are observed as the value of the Fo (minimal fluorescence level when plastoquinone electron acceptor pool, Qa, is fully oxidized) and the ratio of Fv and Fm (maximum quantum efficiency of photosystem-II), which are primarily sensitive to environmental stress conditions [10,58]. AR in tomato seedlings increases Fo value and decreases Fv/Fm ratio, which indicate that AR stress can knock down the photochemical efficiency of PSII photosynthetic apparatus [57,58]. Similarly, AR decreased the efficiency of photosystem II in different crops, like maize and amaranth seedlings [43]. The possible reason was that the high acidity level of AR water can damage the photosynthetic pigments and injure the assimilation tissues.

In addition, photosynthetic pigments in plants may be degraded under AR stress because of the disruption of chloroplasts and lowering of water potential in the cells, which further lead to closure of stomata and lower CO_2_ assimilation, subsequently resulting in introverted cell division [17,63]. In Figure 1, the ultrastructure of a mesophyll cell and chloroplast in the midrib of tomato leaves, observed by transmission electron microscopy (TEM), showed asymmetrical size and shape of chloroplasts, starch grana and a recognizable cell wall without the perfect shape of a cell under the AR stress condition [57]. Likewise, the distorted lamellar structure of chloroplasts and the dwindling chloroplasts collapsed with imperfect thylakoid structure were found in tomato leaf ultrastructure after AR treatment [57]. The perfect starch grana, thylakoids and lamellar structure in leaves directed to development of photosynthetic pigments and enhancement of light energy absorptions and transformation of light energy capacity [16]. AR stress condition with lower pH level disrupts the chloroplast structure and thylakoids, and sometimes the starch grana disappears, which might block the photosynthetic transport in leaves and ultimately inhibits photosynthesis in plants [9,16].

## 3. Reactive Oxygen Species (ROS) Stress and Its Metabolic Mechanism

Plants are immobile in nature, but when any plant faces any adverse environmental condition, they can transform their own physiological status to adapt to an unfavorable environment. In any plant exposed to a harsh growing condition, a rapid and significant disparity ascends within the plant cells to survive. Plants in any stress condition result in the production of huge extents of ROS in mitochondria, chloroplasts and peroxisomes [64]. ROS accumulation can cause abnormalities to these organelles [65], by oxidizing proteins, lipids and nucleic acids [66]. Hydrogen peroxide (H_2_O_2_) acts as one of the utmost active, deadly and damaging ROS. Hydrogen peroxide performs dual role in plants. H_2_O_2_ at low concentrations acts as a signaling molecule, boosting tolerance to any stresses, whereas, at high concentrations, it leads to plant cell death due to oxidative damage [67]. It has been reported that a high concentration of H_2_O_2_ in the plant cells often causes oxidative stress, which finally breaks the antioxidant level, resulting in leaf senescence and sometimes death [44,61]. It has universally been stated that AR with high concentration of acidity markedly accelerates the accumulation of ROS by generating more H_2_O_2_ [8,17]. Similarly, AR stress causes enhanced production of H_2_O_2_ in leaves of plants, which is linked to the acidity extent of AR water [57,58].

On the other hand, due to accumulation of excess ROS and resultant redox imbalance, lipids peroxidation, the damaging process occurs inevitably in every living cell. Sometimes, membrane damage is considered as a sole index to evaluate the extent of lipid destruction under various environmental stresses. It was established that during lipid peroxidation, products are formed from polyunsaturated precursors, including small hydrocarbon fragments like ketones and MDA [68]. MDA acts as a distinctive constituent of reactive carbonyl species produced as a result of lipid peroxidation, and it is considered as a bio-indicator of free radical-catalyzed peroxidation [69,70]. This MDA forms colored thiobarbituric acid-reactive substances (TBARS) by reacting with thiobarbituric acid (TBA) [71]. Lipid peroxidation (MDA) occurs in both cellular and organelle membranes when the threshold limit of ROS is exceeded. As a result, the produced MDA not only exerts a direct effect on normal cellular function, but also aggravates the oxidative injury through the function of lipid-derived radicals [72]. AR stress can encourage membrane injury, and increase permeability of membrane and the buildup of free radicals in plants [61]. Several experiments have been performed, observing the alteration of MDA concentration under AR stress conditions in leaves of different plants, including *Arabidopsis* [73], soybean [74], rice [75] and *Horsfieldia hainanensis* [76]. Likewise, it was reported that tomato seedlings exposed to AR considerably increased the accumulation of MDA [57,58].

Therefore, diverse bio-regulators are stimulated in the amendment to the adverse environment to boost the predominant competences of bioremediation. Notably, plants activate their antioxidant defense system to save themselves from the injurious impacts of ROS in AR stress conditions, alike to other harsh environments [12,13]. In addition, both the enzymatic and non-enzymatic antioxidant compounds act as an antioxidant system to balance ROS and cell membrane stability in stress conditions [77]. The superoxide dismutase (SOD) is plentiful in most of the aerobic organisms and all subcellular compartments, and is considered as a very effective intracellular enzymatic antioxidant which is susceptible to ROS-intervened oxidative stress in different environmental stress [44,61]. Tomato plant exposed to AR stress changes the activity of SOD. Moderate AR with pH 3.5 or more may increase the activity of SOD in tomato seedlings, but severe AR conditions having pH 2.5 might break the SOD activity power in tomato plants [57,58]. Similarly, ascorbate peroxide (APx) is assumed to perform a crucial role in protecting cells in higher plants and other organisms through quenching ROS under adverse environments. APx is involved in ROS detoxification in water–water and ascorbate–glutathione cycles, and also in making use of ascorbate as the electron donor [78]. It was observed that the expression of APx changed markedly under AR stress conditions in leaves of tomato plants, where the rate of alteration differed with the acidity levels of AR water [57,58]. APx can be differentiated from peroxidase (POD) in plants in terms of variances in sequences and physiological activities. POD transforms to indole-3-acetic acid (IAA) and plays a role in lignin biosynthesis and in defense against different stresses via overriding hydrogen peroxide. Peroxidase desires aromatic electron donors, for example guaiacol and pyragallol, which typically oxidize ascorbate at about 1% the rate of guaiacol [79]. The peroxidase activity in plants noticeably depends on plant species and stress condition. Several researchers observed that POD activity in plants significantly increased with AR treatment compared to normal condition [8,12]. Similar to other plants, AR treatment increased POD activity in tomato plants, and the rate of POD activity enhancement depends on plant species and acidity level in AR water [57,58]. In addition, catalase (CAT) is tetrameric heme comprising antioxidant enzymes having the capacity to convert H_2_O_2_ into H_2_O and O_2_. CAT plays a vital role in detoxifying ROS under any abiotic stress conditions [68]. Like other antioxidant enzymes, CAT activity was also found to be increased in tomato seedlings under AR stress conditions depending on the species and level of pH in AR water [57,58].

On the other hand, there are some important non-enzymatic enzymes, e.g., phenolic, flavonoid and proline, that act as ROS scavengers in plants under different environmental stress conditions. Those non-enzymatic antioxidants perform a significant role in the cell structure and function, maintaining the redox status of cells [80]. Like other abiotic stresses, AR stress in plants increases the accumulation of phenolic, flavonoid and proline, and these phenolic, flavonoid and proline help in indirect ROS scavenging, intracellular redox-homeostasis rebuilding and advancement of cellular signaling [44,56,81]. It was observed that the non-enzymatic activity including phenolic, flavonoids and proline increased in tomato seedlings under AR stress [57,58]. But, it was also observed that these enzymatic and non-enzymatic activities could not compensate for the damages caused by severe AR stress in tomato plants [58]. From the above discussion, it can be summarized that enzymatic and non-enzymatic antioxidant activities might help in balancing ROS accumulation and detoxification in plants when the plants are exposed to mild AR stress conditions, but the detoxification capacity can be broken in severe AR stress conditions (Figure 2).

## 4. Mechanism of Acid Rain in Related Gene Expression in Plants

The physiological and biochemical activities, including photosynthesis, generation of ROS, enzymatic and non-enzymatic antioxidant defense, alternation of plant secondary metabolites and stress responsive transcriptional factors in plants, under any abiotic stress conditions, including acid rain stress, can be confirmed by their gene expression pattern [15,18,82,83,84]. It was observed that a series of genes are engaged in the photosynthesis system of *Arabidopsis thaliana*, for example, At2g01590 and At4g27880 genes are known as photosynthetic electron transport chain-related genes, and At2g34430, At2g05070, At3g08940, At3g27690, At1g15820, At5g54270 and At1g03130 are known as the PSI and PSII constituent protein-related genes that show suppression or expression of downregulation due to AR stress [15]. In contrast, it was observed that the expression of RuBP, known as ribulose-1,5-bis-phosphate, and RuBisCO, known as carboxylase/oxygenase, were evidently dropped in *Arabidopsis thaliana* by AR stress [85]. The proteomic study also stated that the expression of carbonic anhydrase gene transformed markedly, indicating that the photosynthesis is vulnerable to some extent against AR [85]. ROS formation in plant cells due to AR stress can be witnessed by the accumulation of superoxide anion (O^−2^) and H_2_O_2_, which triggers membrane damage through producing MDA. Plants try to recover the ROS-induced membrane damage by altering their antioxidant components. Therefore, different enzymatic and non-enzymatic antioxidant activities in plants under environmental stress conditions can be well-observed by the expression pattern of their related genes, for example, CAT1 for catalase, Mn-SOD, Fe-SOD and Cu/Zn-SOD for superoxide dismutase, POD1 for peroxidase, APx genes for L-ascorbate peroxidase, GST genes for glutathione S transferase and AA genes for ascorbic acid were observed to be changed under AR condition [8,12,17]. Furthermore, in earlier studies, some genes were reported to be induced by AR stress conditions in plants, which were directly involved in the ROS-scavenging pathway, such as At1g08830 and At4g25100, documented as superoxide dismutase genes, At4g11600 and At2g25080, recognized as glutathione peroxidase genes, At3g49120, known as a class III peroxidase gene, At5g03630, identified as a monodehydroascorbate reductase gene, At4g35090, isolated as a peroxisomal catalase gene, and At5g16400, At1g07700 and At1g08570, accepted as thio-redoxins genes [15].

A transcriptome study in tomato seedlings showed that 182 differentially expressed genes (DEGs) were upregulated but 1046 DEGs were downregulated under AR stress condition [18]. Gene ontology (GO) analysis of this study showed that 28.86% of DEGs were involved in biological process, 46.34% of DEGs were involved in cellular process and 24.81% DEGs were involved in molecular function in AR-stressed tomato seedlings in comparison to controlled tomato seedlings. In addition, a significant number of DEGs, 174, were found to be associated with biosynthesis pathways of secondary metabolites, including phenylpropanoid, flavonoid, stilbenoid, diarylheptanoid and gingerol, phenylalanine metabolism, starch and sucrose metabolism, amino sugar and nucleotide sugar metabolism, cutin, suberine and wax biosynthesis, metabolic pathways, limonene and pinene degradation, ubiquinone and other terpenoid-quinone biosynthesis, flavone and flavonol biosynthesis, anthocyanin biosynthesis, brassinosteroid biosynthesis, zeatin biosynthesis, arginine and proline metabolism, cysteine and methionine metabolism and carotenoid biosynthesis with respect to AR stress versus control plants through KEGG (Kyoto Encyclopedia of Genes and Genomes) analysis, and among them, only 25 genes were upregulated and the other 149 genes were downregulated [18].

On the other hand, expression of transcriptional factor (TF) family genes is involved in different mechanisms in response to environmental stresses in plants. It has been well-established in several comparative transcriptome analyses that over 30 TF family genes, including *MYB*, *WRKY*, *ERF* and *bZIP*, were found under different environmental stresses [86,87]. The TF genes like *MYB*, *WRKY*, *ERF* and *bZIP* act as fundamental regulators in abiotic stress signal transduction as well as complex in biosynthesis of plant secondary metabolites in response to harsh environments [88,89,90]. Liu et al. [15] observed that AR treatment induced MYB transcription factor, zinc finger proteins, WRKY transcription factors and calcium signal pathway-related genes in plants. In another study, Debnath et al. [18] showed that 151 TF genes which were associated with 31 types of transcript factor family protein were expressed in comparison with control and AR-stressed tomato seedlings, and among the TF-allied DEGs, different stress responsive genes such as *ERF*, *MYB*, *WRKY*, *NAC*, *bHLH, TCP*, *G2-like* and *C2H2* family protein-related genes were markedly expressed. These findings suggested that genes associated with biosynthesis of secondary metabolites and transcriptional factors are significantly induced by AR stress conditions in tomato plants.

## 5. Impact of Silicon in Plants under Acid Rain Stress Conditions

Silicon is well-known as a useful element which can improve the biotic and abiotic stress tolerance in plants [91]. The silicon content in plants varied in between cultivars and species plant growth stages [92]. It is well-established that silicon can advance the chlorophyll content and photosynthesis rate through adjusting oxidative damage in different environmental stresses, like high temperature, salinity, drought and heavy metal [93,94,95,96,97,98]. In addition, Ju et al. [16] observed that the application of exogenous silicon in plants improved the chloroplast ultrastructure, chlorophyll content and rate of photosynthesis, as well as plant growth under moderate to severe AR stress. The integration of silicon in plants under AR stress conditions can improve stomatal conductance and lessen the intercellular CO_2_ concentration, which results in the stimulation of photosynthesis and growth in plants [99]. AR stress tolerance in plants depends on the concentration of silicon and the level of pH in AR water [16]. Likewise, the application of silicon in tomato seedlings could increase the photosynthesis pigment, including chlorophyll and carotenoids, as well as improve the growth under abiotic and biotic stress conditions [100,101]. Moreover, the chlorophyll fluorescence parameters such as Fv/Fm rate (maximum photochemical efficiency of PSII), ETR (photosynthetic electron transport rate) and q_p_ (photochemical quenching coefficient) were increased by silicon supplementation in tomato seedlings under stress conditions [16,100]. It was also observed that the application of silicon in tomato seedlings under stress conditions upregulated the expressions of photosynthesis-related genes such as *PsbP*, *PsbQ*, *PsbW*, *Psb28*, *PetE* and *PetF* [100]. Furthermore, silicon can alleviate stress in plants by increasing the activity of ROS-scavenging antioxidant compounds such as SOD, CAT, POD, phenylalanine ammonialyase (PAL) and polyphenol oxidase (PPO) in plants [22,23,102]. These results demonstrate that exogenous silicon might improve the AR stress tolerance by stimulating the physiological and biochemical activities in tomato plants.

## 6. Effect of Glutathione in Plants under Acid Rain Stress

Glutathione is a low molecular thiol tripeptide compound universally distributed in all the subcellular organelles of plants and plays a crucial role in life processes by removing cytotoxic hydro-peroxides and free radicals, maintaining the thiol level in proteins exchanging thiodusulfide and amino acid transportation across the cell membranes [24,103]. Glutathione has a remarkable role in plant growth, development and response to stresses [103]. It triggers cellular defense against ROS in plants under abiotic stress conditions due to its redox and nucleophilic properties [24,80]. The high concentration of glutathione in the cells regulates a buffering system to maintain redox-balance [104]. Glutathione can quench free radicals directly or by accompanying ascorbic acid in the ascorbate–glutathione cycle which helps in defending cell components from stress-induced oxidative damage [25]. It has been established that supplementation of glutathione improves the plants’ tolerance to stress, such as tolerance in rice to salinity [105], tolerance in tomato to cadmium [104], tolerance in wheat to lead [24] and tolerance in fenugreek to AR [17].

The application of glutathione markedly improves different growth parameters such as plant height and fresh and dry biomass of root and shoot under AR stress through minimizing ROS accumulation and improving the activities of antioxidant enzymes that eventually reduce oxidative stress [17]. It has been reported that most of the plants could activate their antioxidant system up to a certain level against abiotic stress conditions [106,107]. However, the supplementation of glutathione in AR-stressed plants causes further intensification in the antioxidant activities, possibly owing to the signaling role of applied molecules, and this was confirmed through the transcript profiling of CAT, *Mn-SOD*, *Fe-SOD* and *Cu/Zn-SOD* genes [17]. In addition, glutathione-induced enhancement in abiotic stress tolerance in plants was closely associated with the upregulation of numerous transcriptional factors, including ERF (ethylene responsive transcriptional factor), MYB transcriptional factor and other stress response genes [104]. Therefore, it can be suggested that exogenous glutathione acts as an important bio-stimulator in avoiding oxidative damages by activating their defensive genes, which results in the improvement of stress tolerance in the plants against harsh environments [17,24,104].

## 7. Role of Melatonin in Plants under Acid Rain Stress

Melatonin is naturally available in all types of plants and its biosynthesis leads to the significant mechanisms in plants for the survival against different stresses [47,108,109]. Melatonin directly contributes to the plant defense through scavenging free radicals and thus mitigates abiotic and biotic stresses [38,40]. Melatonin also indirectly enhances plant tolerance through recovering leaf ultrastructure, improving the photosynthesis system and regulating plant growth regulators [110]. In this context, exogenous melatonin showed amazing mechanisms to cope with the adverse environments by facilitating plant growth regulation, decelerating leaf senescence, improving photosynthesis and increasing ROS quenching antioxidant systems in plants [111,112]. However, melatonin-mediated physiological and molecular activities in plants prove that melatonin is an efficient molecule to stimulate plant growth, particularly where environmental stresses are the limiting factors for crop production.

Similarly, as observed in Reference [57], the foliar application of exogenous melatonin considerably increased antioxidant activities, reduced ROS and lipid peroxidation and thereby improved the growth, photosynthesis and leaf ultrastructure, indicating AR stress tolerance in tomato plants. An earlier study showed that less damage to chloroplast and comparatively thicker leaf tissues were observed in melatonin-treated plants with respect to the stressed plants [111]. In addition, the mitigation of chlorophyll degradation and improvement in photosynthesis of plants was observed by the application of exogenous melatonin under abiotic stress [113,114]. Numerous previous studies also revealed that exogenous melatonin has complex and influential effects on scavenging ROS, activating antioxidant enzymes and non-enzymes under various harsh conditions [50,115,116,117,118]. Debnath et al. [57] found 100 µM melatonin treatment as the most effective dose among different used doses in AR-stressed tomato plants.

In addition, other experimental results [119] exhibited that SAR-treated tomato plants had improved activities of enzymatic antioxidants in tomato as well as high amounts of health-promoting bioactive compounds in fruits, whereas tomato production greatly decreased. The stress-induced enzymatic antioxidants in tomato might play a significant role to accelerate protection mechanisms by attenuating oxidative stress under different environmental stress conditions, which results in improved biochemical properties of tomato, but is unable to hinder the detrimental effects of stress to yields [120,121,122,123,124]. In contrast, the supplementation of melatonin in AR-stressed plants showed more augmentation of the enzymatic antioxidants and different bioactive compounds in fruits, as well as a sharp increase of the yield attributes of tomatoes [119]. Consistently, it was documented that melatonin enhanced fruit quality during the developmental and ripening stages by reducing degradation of the cell wall and intercellular adhesion [125]. Moreover, these results complied with the concept of other researchers who state that the application of melatonin boosted the stress tolerance of plants by uplifting ROS-detoxifying antioxidants against oxidative injury, improving crop yield under abiotic stress conditions [47,126]. In a recent study, Debnath et al. [119] revealed that melatonin supplementation could mitigate the negative impact of AR stress on tomato fruits by strengthening the antioxidant system and also by increasing health-promoting antioxidant compounds in fruits, and eventually, increase the yield.

It was reported that DEGs were influenced by AR stress in tomato plants, and foliar spray of melatonin in AR-stressed plants showed remarkable expression of DEGs to improve AR stress tolerance in tomato plants [18]. Their results [18] of RNA-sequence and qRT-PCR suggested that the regulatory genes of different secondary metabolites were downregulated by AR treatment in relation to control vs AR-stressed tomato plant (Table 1). Similar to these results, Liu et al. [127] observed that AR treatment changed the expression pattern of differential genes and also the genes associated with secondary metabolites in plants. To the contrary, the use of melatonin in AR-stressed plants caused upregulation of secondary gene expression in control vs melatonin-treated AR-stressed plants, and AR-stressed plants vs melatonin-treated AR-stressed plants (Table 1) [18]. In support of these outcomes, many other experimental outputs also reported that exogenous melatonin upregulates the expression of genes linked with different metabolites and thus, enhances stress tolerance in plants [128,129,130]. In addition, transcriptome analysis also reported that more than 30 TF family genes are involved in stress signal transduction and biosynthesis of plant secondary metabolites under abiotic stress [88,89,90]. According to qRT-PCR and RNA-sequence results of the study of Debnath et al. [18], it was stated that the expression of MYB, WRKY, ERF and bZIP were down regulated by AR treatment in relation to control (Table 1). Liu et al. [15] also observed the down regulation of TF-related genes in *Arabidopsis* under AR stress conditions. Conversely, the results of Reference [18] also revealed that the foliar spray of melatonin in AR-stressed plants enhanced the upregulation of stress-responsive TF-related genes to lighten SAR stress. It has been perceived that application of melatonin adjusts the expression of TFs such as bZIP, MYB, WRKY and ERF, which accelerates the expression of ROS-scavenging enzyme-encoding genes to promote abiotic stress tolerance [131,132,133]. However, Debnath et al. [18] exposed the genes associated with the activation of antioxidants, modulation of secondary metabolites and their pattern of expression to melatonin treatment under AR stress.

Therefore, the series of experimental results [18,57,119] reported that the use of melatonin might be a potential technique for enriching plant tolerance by modulating growth, physiological and molecular activities in AR condition. Figure 3 presents a summary showing how exogenous melatonin influences cellular mechanism of AR stress tolerance in plants. Melatonin can freely penetrate the cell membranes because of its amphiphilic nature. Melatonin directly quenches ROS and also upsurges the extent of antioxidant activity to accelerate ROS scavenging capacity, thereby defending cellular damage and improving AR stress tolerance.

## 8. Conclusions and Future Prospective

The current review was carried out to understand the responses of plants to AR stress and elucidate the possible impact of silicon, glutathione and melatonin in mediating AR stress tolerance. The major findings indicated that AR stress reduces the normal plant growth and photosynthesis by stimulating ROS generation and inhibiting subsequent pathways. While the ROS detoxification system was found to be effectively activated and improved normal plant growth and productivity by the application of bio-stimulators under such circumstances. Hence, the stimulation of natural biosynthesis and/or exogenous supplementation of the above-mentioned bio-regulators might establish a new state of equilibrium to contribute to the inherent plasticity of plants in order to combat and detoxify the stress generated by AR. Numerous articles have explored that foliar application of silicon, glutathione and melatonin effectively considerably ameliorates the toxicity of AR in many plant species by mitigating the growth, photosynthetic inhibitors, leaf ultra-structural changes and antioxidant activities in plants during AR stress (Figure 4). Besides, melatonin applications during AR stress enhanced not only the quality traits but also the bioactive antioxidant compounds in fruits, which has enormous health benefits. Furthermore, the current review also summarized that in addition to physiological responses, the protective roles of exogenous silicon, glutathione and melatonin to combat the AR stress are highly involved in the regulation of transcription factors like bZIP, MYB, WRKY and ERF in plants.

Therefore, recent advances of the physiological and molecular activities of silicon, glutathione and melatonin in plants have confirmed their prime roles to improve AR stress tolerance. However, the genetic evidence and the subsequent signaling cascades in the action of AR stress tolerance have not yet been deeply studied, and thus require further investigations. Hopefully, additional underlying mechanisms and core pathways to explain the high efficiency of the argued bioactive compounds in enhancing the tolerance of plants to AR stress will be uncovered in the near future. Hence, the adoption of new tactics in mitigating the consequences of AR stress on crops’ growth and productivity have promise in escalating agricultural extension, where AR is a restrictive factor.

## Figures and Tables

**Figure 1 molecules-26-00862-f001:**
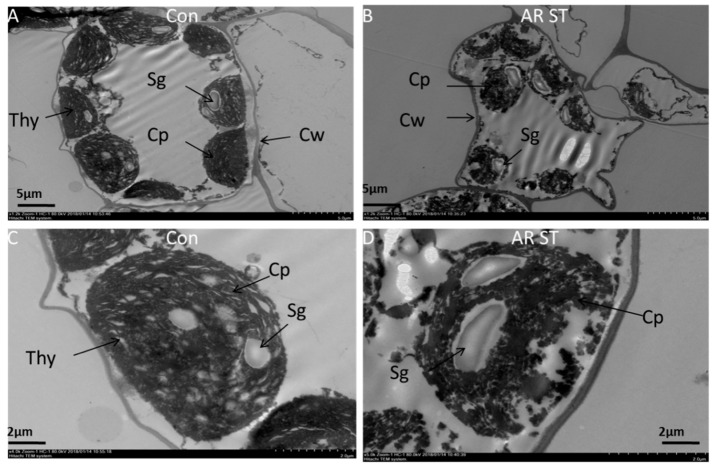
Ultra-structures of mesophyll cells of the middle part of midrib of the tomato leaf through transmission electron microscopy (TEM). (**A**,**B**) figures show TEM structure of whole leaf mesophyll cell of control and acid rain (AR)-stressed (AR-ST) tomato leaf, respectively. (**C**,**D**) figures show a relatively low magnified view of mesophyll cell of control (Con) and AR-stressed (AR-ST) tomato leaf, respectively. CP, Chloroplast; CW, Cell wall; Thy, Thylakoids; Sg, Starch grana [57].

**Figure 2 molecules-26-00862-f002:**
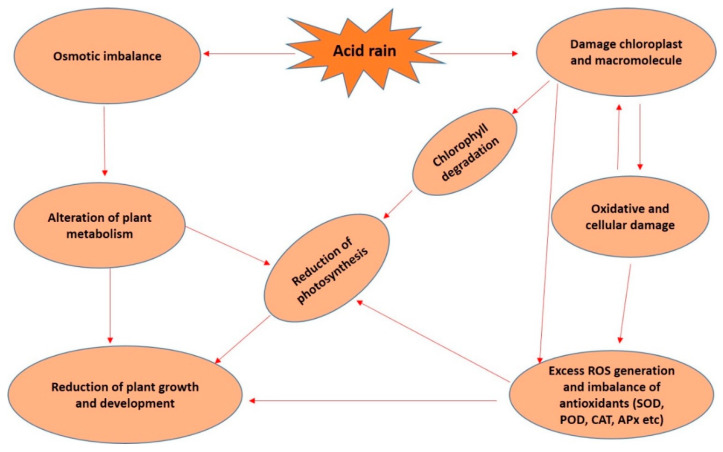
The action of acid rain stress on photosynthesis, antioxidant system and growth in plants. Here, ROS—reactive oxygen species, SOD—superoxide dismutase, POD—peroxide, CAT—catalase, APx—ascorbate peroxidase.

**Figure 3 molecules-26-00862-f003:**
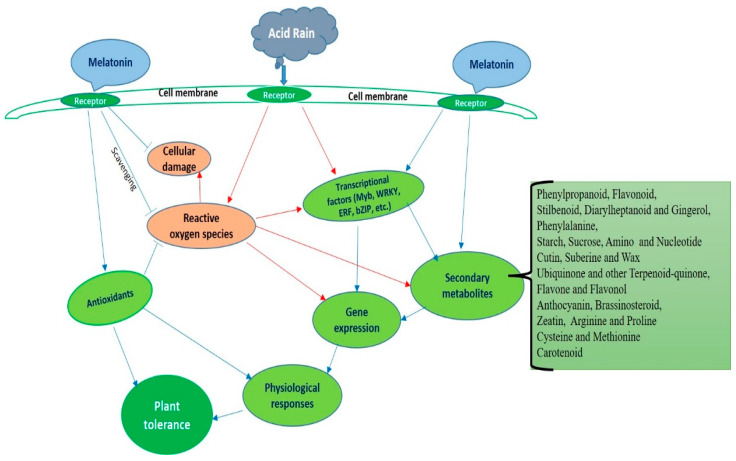
Melatonin-mediated acid rain stress tolerance and detoxification mechanism in plants. Modified from Debnath et al. [18].

**Figure 4 molecules-26-00862-f004:**
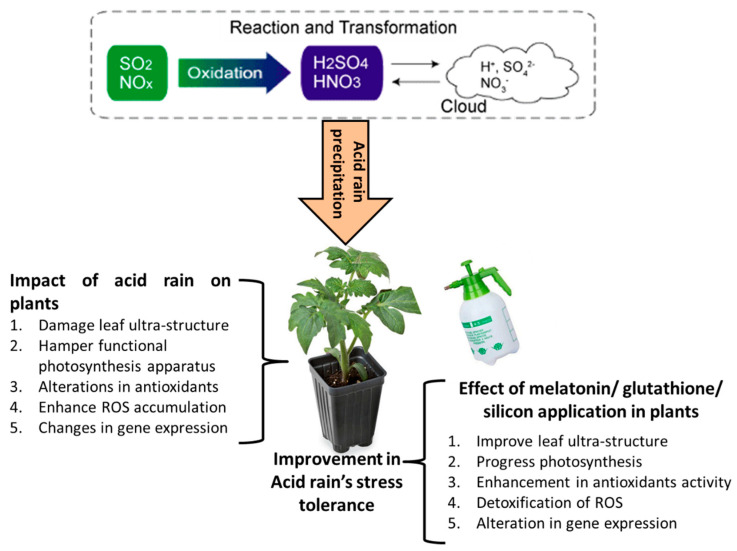
The role of melatonin, glutathione and silicon in mitigating the acid rain effects in plants.

**Table 1 molecules-26-00862-t001:** Expression of differentially expressed genes (DEGs) through RNA sequence (log2 fold change) and qRT-PCR fold changes in tomato plant under AR stress conditions and melatonin supplementation [18].

Gene Description	Function	Expression	
		Control vs. AR-Stressed Plants	AR-Stressed Plants vs. AR-Stressed Plants Treated with Melatonin
Caffeoyl-CoA *O*-methyltransferase-like	Biosynthesis of secondary metabolites, Phenylpropanoid biosynthesis, Flavonoid biosynthesis, Stilbenoid, diarylheptanoid and gingerol biosynthesis, Phenylalanine metabolism, Metabolic pathways	Downregulated	Upregulated
Probable galacturonosyl transferase-like 10-like	Biosynthesis of secondary metabolites, Starch and sucrose metabolism, Amino sugar and nucleotide sugar metabolism	Downregulated	Upregulated
Cytochrome P450 94A1-like	Biosynthesis of secondary metabolites, Cutin, suberine and wax biosynthesis, Stilbenoid, diarylheptanoid and gingerol biosynthesis, Metabolic pathways, Limonene and pinene degradation	Downregulated	Upregulated
Salutaridinol 7-*O*-acetyltransferase-like	Biosynthesis of secondary metabolites, Phenylpropanoid biosynthesis, Flavonoid biosynthesis, Stilbenoid, diarylheptanoid and gingerol biosynthesis	Downregulated	Upregulated
4-coumarate-CoA ligase 2-like	Biosynthesis of secondary metabolites, Phenylpropanoid biosynthesis, Phenylalanine metabolism, Metabolic pathways, Ubiquinone and other terpenoid-quinone biosynthesis	Downregulated	Upregulated
Anthocyanidin 3-*O*-glucosyltransferase-like	Biosynthesis of secondary metabolites, Metabolic pathways, Flavone and flavonol biosynthesis, Anthocyanin biosynthesis	Downregulated	Downregulated
Secologanin synthase-like isoform 1	Biosynthesis of secondary metabolites, Metabolic pathways, Brassinosteroid biosynthesis, Zeatin biosynthesis	Downregulated	Upregulated
Caffeoyl-CoA *O*-methyl transferase-like isoform 2	Biosynthesis of secondary metabolites, Phenylpropanoid biosynthesis, Flavonoid biosynthesis, Stilbenoid, diarylheptanoid and gingerol biosynthesis, Phenylalanine metabolism, Metabolic pathways	Downregulated	Downregulated
S-adenosylmethionine decarboxylase proenzyme-like	Arginine and proline metabolism, Cysteine and methionine metabolism, Metabolic pathways	Downregulated	Upregulated
Cyanidin-3-*O*-glucoside 2-*O*-glucuronosyltransferase-like	Flavone and flavonol biosynthesis, Zeatin biosynthesis	Downregulated	Upregulated
Abscisic acid 8′-hydroxylase 1-like	Carotenoid biosynthesis	Downregulated	Upregulated
MYB-related protein Myb4-like	Stress-responsive MYB family transcriptional factor	Downregulated	Upregulated
Probable WRKY transcription factor 33-like	Stress-responsive WRKY family transcriptional factor Plant–pathogen interaction	Downregulated	Upregulated
Ethylene-responsive transcription factor 1-like	Stress-responsive ERF family transcriptional factors Plant hormone signal transduction	Downregulated	Upregulated
Uncharacterized protein LOC101262884 isoform 1	Stress-responsive bZIP family transcriptional factor	Downregulated	Upregulated

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
