# Peer review of "Physiological and Molecular Responses to Acid Rain Stress in Plants and the Impact of Melatonin, Glutathione and Silicon in the Amendment of Plant Acid Rain Stress"

_molecules, 2021, doi:10.3390/molecules26040862_

Round 1
Reviewer 1 Report
Remarks are included in the manuscript.

Author Response
Author’s Response to Reviewer-1 Comments
Manuscript ID:molecules-1085953
Manuscript Title: Physiological and Molecular Responses to Acid Rain Stress in Plants and Impact of Melatonin, Glutathione and Silicon in the Amendment of Plant Acid Rain Stress
The authors would like to thank the Reviewer-1 for his deep review of our manuscript with very constructive comments and suggestions. The general and specific questions or concerns and recommendations are highly appreciated, and corrections have been made accordingly in the revised version of the manuscript to improve its quality. We have done necessary correction according to kind remarks in manuscript. The point-by-point responses to the reviewer’s comments and suggestions are listed as follows:
Comment-1: The role of phytochelatins should be added
Response-1: Added according to your kind suggestion. Please see the line 69-71.
Comment-2: L84-86 should be rewritten and highlight sentences were repeat with L84-6 sentence. Rewrite this paragraph.
Response-2: Rewritten this paragraph were done according to your kind suggestion. Please see the line 86-94.
Comment-3: Remove Bar = 5µm and Bar = 2µm. Bars are explained in figure 1
Response-3: Removed from figure 1 legend. Thank you so much for your kind suggestion.
Comment-4: Explain ASH-GSH abbr.
Response-4: Full name were mentioned according to your kind suggestion. Please see the line 188-189.
Comment-5: Acid rain helps or phenolic, flavonoid and proline help?
Response-5: Corrected in line 205-206. Thank you so much for your deep review.
Comment-6: couldn’t without abbr.
Response-6: Full form mentioned. Please see the line 209. Thanks
Comment-7: I suggest prepare summarize diagram which shows silicon, glutathione and melatonin role in mitigating the acid rain effects in plants
Response-7: Prepared summarize diagram in figure 4 in conclusion section according to your kind suggestion. Thank you so much for your kind suggestion.
Comment-8: Conclusion and future prospective section is unclear. Rewrite
Response-8: Conclusion and future prospective section were rewritten. Thank you so much for your kind improvements.
We are thankful you for your kind review and favorable consideration.
With Thanks
Dongliang Qiu
Reviewer 2 Report
It was a pleasure to read the manuscript by Debnath and colleagues.
The authors made a complete and current analysis of the ecological effects of acid rain on plants. Furthermore, some molecular mechanisms involved in this abiotic stress have been hypothesized.
The manuscript is clear and complete, however some typos are present in the text and should be corrected. I also suggest the inclusion of two manuscripts in the references:
Reza Yousefi, A.; Rashidi, S.; Moradi, P.; Mastinu, A. Germination and Seedling Growth Responses of Zygophyllum fabago, Salsola kali L. and Atriplex canescens to PEG-Induced Drought Stress. Environments 2020, 7, 107.
Rad, S.V.; Valadabadi, S.A.R.; Pouryousef, M.; Saifzadeh, S.; Zakrin, H.R.; Mastinu, A. Quantitative and Qualitative Evaluation of Sorghum bicolor L. under Intercropping with Legumes and Different Weed Control Methods. Horticulturae 2020, 6, 78.
Author Response
Author’s Response to Reviewer-2 Comments
Manuscript ID:molecules-1085953
Manuscript Title: Physiological and Molecular Responses to Acid Rain Stress in Plants and Impact of Melatonin, Glutathione and Silicon in the Amendment of Plant Acid Rain Stress
The authors would like to thank the Reviewer-2 for his deep review of our manuscript with very constructive comments and suggestions. The general and specific questions or concerns and recommendations are highly appreciated, and corrections have been made accordingly in the revised version of the manuscript to improve its quality. The point-by-point responses to the reviewer’s comments and suggestions are listed as follows:
Comments: It was a pleasure to read the manuscript by Debnath and colleagues.
The authors made a complete and current analysis of the ecological effects of acid rain on plants. Furthermore, some molecular mechanisms involved in this abiotic stress have been hypothesized.
The manuscript is clear and complete, however some typos are present in the text and should be corrected. I also suggest the inclusion of two manuscripts in the references:
Reza Yousefi, A.; Rashidi, S.; Moradi, P.; Mastinu, A. Germination and Seedling Growth Responses of Zygophyllum fabago, Salsola kali L. and Atriplex canescens to PEG-Induced Drought Stress. Environments 2020, 7, 107.
Rad, S.V.; Valadabadi, S.A.R.; Pouryousef, M.; Saifzadeh, S.; Zakrin, H.R.; Mastinu, A. Quantitative and Qualitative Evaluation of Sorghum bicolor L. under Intercropping with Legumes and Different Weed Control Methods. Horticulturae 2020, 6, 78.
Responses: We are really grateful for your kind consideration and valuable recommendation to improve the quality of our manuscript. Some typos were corrected in the text according to your kind suggestion. We have included one reference which one related to our manuscript (Reza Yousefi, A.; Rashidi, S.; Moradi, P.; Mastinu, A. Germination and Seedling Growth Responses of Zygophyllum fabago, Salsola kali L. and Atriplex canescens to PEG-Induced Drought Stress. Environments 2020, 7, 107) according to your kind recommendation. Please see the reference number 45 and the line 109. We could not found any suitable place in the manuscript to cite another reference. We are really sorry for that.
We are thankful you for your kind review and favorable consideration.
With Thanks
Dongliang Qiu
Round 2
Reviewer 1 Report
A revised version of this manuscript is well-organized and well-written.